antenatal depression; health workers; identification; screening; task shifting

**Corresponding author:**
Susan Thomas;
Email: susanthomas@sjri.res.in

# Feasibility of training primary healthcare workers to identify antenatal depression

Susan Thomas[1] , Maria Ekstrand[2], Tinku Thomas[3] and Krishnamachari Srinivasan[1,4]

[1]Division of Mental Health and Neurosciences, St. John's Research Institute, Bengaluru, Karnataka, India; [2]Division of Prevention Science, Department of Medicine, University of California, San Francisco, USA; [3]Department of Biostatistics, St. John's Medical College, Bengaluru, Karnataka, India and [4]Department of Psychiatry, St John's Medical College, Bengaluru, Karnataka, India

## Abstract

Identifying women with depressive symptoms is the first step to reducing the risk of the short-term and long-term consequences of antenatal depression. Task shifting by training primary healthcare workers may help to reduce the burden in low-resource settings. Twenty health workers in a primary healthcare center in urban Bengaluru were trained to screen and identify antenatal depression. The training had two components: knowledge-based, using the depression module in the Mental Health Gap Action Program; and skills-based hands-on training, using the Patient Health Questionnaire-9. Knowledge about antenatal depression in the health workers improved by three units after training ($p < 0.001$). Their perceived skills and self-efficacy also improved by one unit each ($p = 0.032$ and $p = 0.036$, respectively). Following the training, 25% of the pregnant women who underwent screening by health workers reported depressive symptoms, as compared to no positive screening before training. Training was found to improve the knowledge, perceived skills and self-efficacy of nurses, junior health assistants and Accredited Social Health Activists (ASHAs), and was found to increase the screening rate of depression in an antenatal clinic in urban India. Incorporating screening for depressive symptoms into regular antenatal care is feasible in low-resource settings.

## Impact statement

Even though pregnancy is considered a period of bliss, several women experience mental health problems during this time, especially depression. The number of pregnant women with depressive symptoms is high in low- and middle-income countries, and as these women are at the prime of their lives, the burden is high for them, their families and society. Depression during pregnancy has a significant adverse impact on their future mental health and on the cognitive and behavioral development and the mental health of their children. The lack of availability of an adequate number of trained professionals and less accessibility to mental healthcare along with the stigma attached to mental health disorders further aggravate this problem. Task sharing by training health workers in primary care facilities can be an effective way to overcome this hurdle. In this study, the feasibility of training health workers from different professions and educational backgrounds to identify depressive symptoms in pregnant women attending an antenatal clinic was examined. It was found that the training not only helped in the identification of depressive symptoms but also improved the health workers' knowledge, perceived skills and self-efficacy. This method can be used in similar settings in other parts of the world to help in decreasing the burden of mental healthcare.

## Social media summary

Training healthcare workers in antenatal clinics to identify antenatal depression is feasible and it can increase the identification rate of depressive symptoms in pregnant women.

## Introduction

Antenatal depression has several adverse short-term and long-term consequences. It has been associated with low birth weight and preterm birth in the short term (Grote et al., 2004; Grigoriadis et al., 2013; Dadi et al., 2020; Fekadu Dadi et al., 2020) and with mental health problems in the mother and child and developmental delays in the children in the long term (Leigh and Milgrom, 2008; Muzik and Borovska, 2010; Raposa et al., 2014). The global burden of

antenatal depression is high with 1 in 10 pregnant women experiencing depressive symptoms globally (Woody et al., 2017; Molenaar et al., 2018). The prevalence of antenatal depression is higher in low- and middle-income countries (LMICs) (Patel et al., 2002; Fisher et al., 2012), with higher cases of disability, more life-threatening events, and poorer family and social support being experienced by depressed mothers as compared to non-depressed mothers (Rahman et al., 2003).

Identifying women with depression is the first step to reducing the risk of adverse outcomes. It is found that incorporating a formal assessment for depression in perinatal care improves treatment engagement and outcome (Miller et al., 2009). A full assessment for screening mood and emotional disorders with a validated instrument is recommended by the American College of Obstetricians and Gynecologists (ACOG) committee ("ACOG Committee Opinion No. 757", 2018). However, the low number of trained mental health professionals (Garg et al., 2019) and lack of time and training of non-mental health professionals (Legere et al., 2017; Mathibe-Neke and Suzan Masitenyane, 2018) in an already burdened health system often results in low rates of identification of perinatal depression in resource-poor settings.

Task shifting by training primary healthcare workers such as nurses, junior health assistants and community link workers or Accredited Social Health Activists (ASHAs) in screening and assessment of depressive symptoms could help in identifying and referring more women with antenatal depression at an early stage. Nurses and other primary healthcare staff can help in the process of identification due to their frequent contact with peripartum women (Segre et al., 2010). A systematic review (Legere et al., 2017) of 12 studies to identify the educational needs of healthcare providers found that providers identified a lack of education in perinatal mental health and the need for further professional development. In a survey conducted across 19 maternity sites in Australia (McCauley et al., 2011) on 160 midwives and nurses, 93% indicated they could be better prepared to provide mental health intervention for women. They felt the need to improve their own and others' skills and knowledge regarding identification of mental health and illness in the peripartum period, and in specific care provision and mental health interventions. Studies that have examined the knowledge and attitude of nurses towards perinatal depression reported fear, anxiety and frustration while dealing with patients with perinatal mental illness (McConachie and Whitford, 2009). In a previous exploratory study conducted by us in a private hospital in Bangalore on the role of nurses in perinatal depression screening and intervention, approximately one-third (32.7%) of the participants reported having frequently encountered a woman who needed mental health evaluation (Thomas et al., manuscript submitted for review). Most of the nurses expressed an interest in perinatal mental health screening and were willing to assume responsibility for identifying depressive symptoms in their patients. However, only half of the nurses surveyed were aware of the screening tools available for perinatal depression (58.4%) (Thomas, et al., manuscript submitted for review).

The present study examined the feasibility of providing training to primary healthcare workers in an urban antenatal clinic in screening and identifying depressive symptoms in pregnant women using a questionnaire. The study also compared the primary healthcare workers' cognitive and behavioral factors related to the identification of antenatal depressive symptoms before and after the training and compared the identification rate of antenatal depression in the clinic before and after training. We hypothesized that the rate of identification of antenatal depression in the PHC would increase and the cognitive and behavioral factors related to the identification of antenatal depression among primary care health workers would improve following training.

## Material and methods

The present study was approved by the Institutional Ethics Committee of the institute and the Health Ministry Screening Committee, Indian Council of Medical Research. The study used an uncontrolled pre–post design. The study was conducted in an Urban Primary Health Center (UPHC) in Bangalore that provides maternity care to low- and middle-income population, including 70,000 urban-slum dwellers. Antenatal check-ups are conducted daily, with approximately 100 new registrations every month.

### Sample

The primary healthcare workers included nurses, junior health assistants and ASHAs working in the UPHC who were invited to participate in the training sessions for screening and identifying antenatal depression. Junior health assistants hold a diploma in Auxiliary Nursing and Midwifery after completing their 10th grade and work as assistants to nurses. ASHAs are trained female community health activists, recruited by the government from the community. They are school-educated and are provided intensive training in healthcare, especially mother and child healthcare for 6 months after recruitment. They work as an interface between the community and healthcare system. 300 healthy women above 18 years, <24 weeks pregnant and fluent in Kannada without a history of mental disorders, serious physical disorders or complications in pregnancy attending the antenatal clinic were screened for depression by the trained staff in the UPHC.

### Tools

#### Survey to assess cognitive and behavioral factors
A 35-item interviewer-administered survey was used to assess the cognitive and behavioral factors related to identifying antenatal depression among non-mental-health primary care workers. Survey items from tools developed for similar studies in India and elsewhere (described below) were modified and used to assess healthcare professionals' cognitive and behavioral factors related to perinatal depression and mental health.

Cognitive factors included the health workers' knowledge (El-Den et al., 2019), expectations about their role in identifying antenatal depression (Thomas et al., manuscript submitted for review) and stigmatizing attitudes toward mental illness (Gabbidon et al., 2013). The behavioral factors assessed the health workers' self-reported skills and their previous experience and practice in the identification of antenatal depressive symptoms.

Knowledge.  Five items measured the participants' understanding of antenatal depression. Items were adapted from a questionnaire measuring basic perinatal depression knowledge (El-Den et al., 2019). These were multiple-choice questions, with one correct answer and three wrong answers. A correct answer got a score of 1, while a wrong answer got a score of 0. The total score is the sum of correct answers and ranged between 1 and 5. A higher score indicates better knowledge. Examples include, 'Antenatal depression refers to depression that arises' with the option to choose the right answer from (A) During pregnancy, (B) Before pregnancy,

(C) After childbirth and (D) None of the above' and 'Which of the following statements is true? with the options (A) Depression in the antenatal period always continues into the postnatal period (B) Antenatal depression can affect women from any cultural, ethnic or religious background (C) Women who are depressed antenatally do not require specific treatment and (D) Antenatal depression is the same thing as pregnancy related mood swings.

Expectations. This section included four items on the health workers' expectations about their own role in the identification of antenatal depression and the ease of doing this. Items from a scale used in a previous study with nurses in India (Thomas et al., manuscript submitted for review) were adapted for use in this study. Examples include, 'If I thought a pregnant woman was depressed, it would be my responsibility to advise her to seek evaluation from a mental health professional' and 'It would be easy to include a 5-min screen for depression/ anxiety in my daily interactions with pregnant women'. The items were scored on a Likert scale as follows: Strongly disagree: 1, Disagree: 2, Do not agree or disagree: 3, Agree: 4 and Strongly agree: 5. The total score is the sum of scores of all individual items. The total score could be between 1 and 20. A higher score meant higher expectations that they had a role to play.

Attitude. These questions assessed the health workers' attitudes toward mental illness. This section included the Attitudes- Mental Illness: Clinicians' Attitudes (MICA-version 4) (Gabbidon et al., 2013). The items measure views of mental illness and psychiatry, knowledge of mental illness, disclosure, distinguishing mental and physical healthcare, and patient care for people with mental illness. The MICA 4 scale is suitable for nurses and other health and social service providers and has been used in previous studies (Desai et al., 2019; Praharaj et al., 2021). The items are scored on a Likert scale. The total score is the sum of the scores for the individual items. For items 3, 9, 10, 11, 12 and 16, items are scored as follows: Strongly agree = 1, Agree = 2, Somewhat agree = 3, Somewhat disagree = 4, Disagree = 5, Strongly disagree = 6. All other items (1, 2, 4, 5, 6, 7, 8, 13, 14, 15) are reverse-scored (Siddiqua and Foster, 2015). The total score ranges between 1 and 96. Higher scores indicate a more negative or stigmatizing attitude toward people with mental illness. Examples include, 'People with severe mental illness can never recover enough to have good quality of life' and 'If I had a mental illness, I would never admit this to my friends because I would fear being treated differently'.

Skills. This section included four items on their perceived skills in identifying depressive symptoms. The responses were scored on a Likert scale. Items were adapted from a scale used in a previous study with nurses (Thomas et al., manuscript submitted for review). The following scores were given to responses: Strongly disagree: 1, Disagree: 2, Do not agree or disagree: 3, Agree: 4, and Strongly agree: 5. The total score is the sum of the scores of all individual items. The total score ranged between 1 and 20. A higher score meant higher perceived skills or comfort. Examples include, 'I am aware of tools that can be used to screen prenatal depression/ anxiety' and 'I am able to identify prenatal depression in pregnant women in the OPD'.

Practice and previous experience. These two items assessed the participants' self-reported experience identifying antenatal depression. Items were adapted from a scale used in a previous study (Thomas et al., manuscript submitted for review). The items are scored on a Likert scale as follows: Rarely: 1, Occasionally: 2, Sometimes: 3, Often: 4, and Very often: 5. The total score is the sum of scores of all individual items. The total score ranged from 1 to 10, with a higher score indicating more practice and previous experience. Examples include, 'How often have you encountered a woman who needs mental health evaluation?' and 'How often have you referred such a patient for mental health evaluation?'

Self-efficacy. This section included four items on the health workers' confidence in their ability to identify antenatal depression. The items are scored on a Likert scale and were adapted from a scale used to measure self-efficacy in the identification of postpartum depression by hospital-based perinatal nurses (Logsdon et al., 2010). The responses are scored as follows: Strongly disagree: 1, Disagree: 2, Do not agree or disagree: 3, Agree: 4, and Strongly agree: 5. The total score is the sum of the scores of all individual items. The total score ranged between 1 and 20. A higher score indicates greater self-efficacy. Examples include 'I am confident in assessing a pregnant woman's basic knowledge of antenatal depression' and 'I am confident in assessing symptoms of antenatal depression in a pregnant woman'.

The survey instrument was translated to Kannada and back-translated to English and was subsequently pilot-tested on 10 nurses to assess applicability in this setting.

### The Patient Health Questionnaire- 9 (PHQ-9)

The Patient Health Questionnaire (PHQ-9) is a nine-item, reliable and valid tool for the identification of depression among pregnant women (Kroenke et al., 2001; Sidebottom et al., 2012). This has been used in previous studies in multiple settings to assess perinatal depression (Wang et al., 2021). It also has an item to identify the presence of suicidal ideation. If suicidal ideation was present, it was further evaluated by the study staff using suicidal assessment questions from the Mini-international neuropsychiatric interview (M.I. N.I) (Sheehan et al., 1998) and referrals to a psychiatrist were made if needed.

### Training of the health workers

The training had two components: knowledge-based and skills-based. The knowledge-based training used the depression module in the Mental Health Gap Action Program (MHGap) manual by the WHO ("mhGAP Intervention Guide - Version 2.0", n.d.). This is an intervention guide for non-specialized settings to scale up services for mental, neurological and substance abuse disorders and includes an introduction to the symptoms and methods for identification using a flow chart. The skills-based training included hands-on training in the administration, scoring and interpretation of the PHQ-9. The training was conducted in small groups of 4–5 participants, and included lectures, discussions, role plays and modeling. Each training session lasted for 3 h with two breaks in between.

### Procedure

The baseline rate of identification of antenatal depressive symptoms was obtained by examining medical records maintained at the UPHC for a three-month period prior to the training. Upon completion of the training, UPHC staff started screening consenting pregnant women who attended the clinic. Women who registered for antenatal check-ups in the clinic were met by our research

staff and were assessed for eligibility in the study. If they were eligible, consent was sought, and the health worker was notified. The woman was accompanied to a quiet room where the screening was done by the health worker, The screening took 10 min. Women scoring between 5 and 9 underwent further evaluation by the research staff and, for ethical reasons, were offered a single session of psycho-education session based on the WHO 'Thinking Healthy' manual (WHO, 2015). The procedure is presented in Figure 1.

### Statistical analysis

Means, SDs and percentages were used to describe the characteristics of study participants. The cognitive factors were summarized as Median (Quartile 1, Quartile 3). Pre–post changes in cognitive and behavioral factors following training were compared using the Wilcoxon sign-rank test. All analyses were performed using the SPSS 17.0 software.

### Results

The study was conducted between January 2020 and March 2021. Twenty primary healthcare workers were recruited for the training on the identification of antenatal depression. 40% of the participants (N = 8) were ASHA workers and 5 (25%) were general nurses, 1 (5%), an obstetric nurse and 6 (30%) were junior health assistants. The mean (SD) age of the health workers was 33.4 (9.7) years and the mean education level was 12.3 (1.9) years (Table 1). Three hundred pregnant women were screened during the study period. There was approximately an equal number of women in their first and second trimesters, and approximately half of them were in their first pregnancy (n = 146, 48.7%). The majority were educated between 8 and 12 years (n = 179, 59.7%), and were self-employed (n = 202, 67.3%). The socio-demographic characteristics of the participating pregnant women are presented in Table 2.

Scores on three of the assessed outcome measures showed improvement after the training. Among cognitive factors, knowledge improved significantly after training. The median (Q1, Q3) was 1 (1,2) before training and 4 (3,4) after training, p < 0.001. Two

**Table 1.** Socio-demographic characteristics of the healthcare workers

| Socio-demographic characteristics of health workers | N = 20 |
|---|---|
| Age (mean years, SD) | 33.4 (9.7) |
| Education (mean years, SD) | 12.3 (1.9) |
| Experience (mean years, SD) | 6.4 (4.9) |
| Designation (N, %) | |
| General Nurse | 5 (25%) |
| Obstetric Nurse | 1 (5%) |
| Junior Health Assistant | 6 (30%) |
| Community link worker | 8 (40%) |

behavioral factors, perceived 'skills' and 'self-efficacy', also improved significantly (p = 0.032 and p = 0.036, respectively). The median (Q1, Q3) of skills was 15 (14,16) before training and 16 (15.19) after training. Median (Q1, Q3) score on self-efficacy was 16 (16,17) before training and 17 (16,19) after training. (Table 3).

The healthcare workers did not identify any pregnant women with depressive symptoms in the PHC in the 3 months prior to the training. However, following the training, UPHC staff identified 25% of the pregnant as screened positive for depressive symptoms on PHQ 9 (Figure 2), with the majority (95%) reporting mild symptoms. The four pregnant women identified with moderate depressive symptoms declined referral to a mental health professional.

### Discussion

This study demonstrated that it is feasible to train nurses, nurse assistants and ASHAs to identify antenatal depression among women presenting to UPHC for routine antenatal visits. The training helped improve the UPHC staff's knowledge, perceived skills and self-efficacy in identifying antenatal depression. The results also showed an increase in the number of women identified with

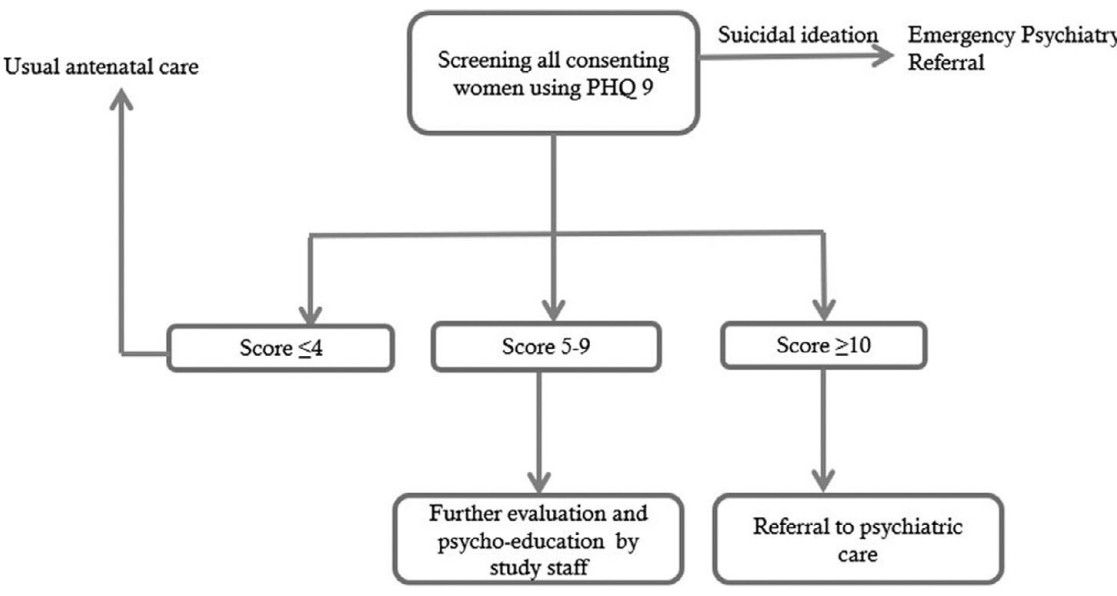

**Figure 1.** Flowchart of the procedure

**Table 2.** Socio-demographic characteristics of the pregnant women who participated in the study

| Socio-demographic characteristics of study participants | N = 300 |
|---|---|
| Age (mean years, SD) | 24.9 (4.2) |
| Trimester | |
| First | 143 (47.6%) |
| Second | 157 (52.3%) |
| Parity | |
| Primiparous | 146 (48.7%) |
| Multiparous | 154 (51.3%) |
| Education (years) | |
| ≤ 8 | 38 (12.7%) |
| 8–12 | 179 (59.7%) |
| > 12 | 83 (27.7%) |
| Family income (mean in INR, SD) | 19,192 (13634) |
| Occupation | |
| Unemployed | 2 (0.7%) |
| Self Employed | 202 (67.3%) |
| Hired workers | 96 (32.0%) |

**Table 3.** Comparison of cognitive and behavioral factors before and after training

| Variable | Pre training assessment Median (Q1, Q3) | Post training assessment Median (Q1, Q3) | *p*-value |
|---|---|---|---|
| Cognitive factors | | | |
| Knowledge | 1 (1,2) | 4 (3,4) | <0.001 |
| Expectations | 17 (16,18) | 18 (16,20) | 0.158 |
| Attitude | 55 (51,58) | 53 (50,55) | 0.053 |
| Behavioral factors | | | |
| Skills | 15 (14,16) | 16 (15.19) | 0.032 |
| Practice and previous experience | 4 (4,7) | 4 (2,5) | 0.084 |
| Self-efficacy | 16 (16,17) | 17 (16,19) | 0.036 |

depressive symptoms once a structured screening was initiated at the UPHC compared to no screening before the onset of the study.

The nurses, nurse assistants and ASHA workers were willing to undergo training to identify antenatal depression. They expressed their interest in attending the training sessions and conducting the screening after the training. This is consistent with findings from previous studies that nurses are interested in identifying mental health problems in the perinatal setting (Thomas et al., manuscript submitted for review) and felt the need to improve their knowledge and skills (McCauley et al., 2011).

The training improved the nurses', junior health assistants' and ASHA workers' knowledge, perceived skills and self-efficacy in identifying depressive symptoms. Researchers have suggested that the fear and anxiety of nurses while dealing with perinatal women

with mental health problems could be reduced by providing them training on mental illness and thus improving their knowledge, skills and experience (McConachie and Whitford, 2009). Since the MHGap intervention guide directly targets the knowledge about depression, this likely contributed to their improvement. The training on PHQ-9 improved the study participants' perceived skills and self-efficacy. Previous studies have noted that nurses reported fear, anxiety and frustration while dealing with patients with perinatal mental illness (McConachie and Whitford, 2009), and expressed a need to improve their skills in identification of mental illnesses (McCauley et al., 2011). Training that helps to improve their perceived skills and self-efficacy could thus help in allaying this fear and make them more prepared for task shifting. This could also help to increase their interest in receiving additional mental health training, as seen in one of our previous studies with ASHAs (Bansal et al., 2021).

There was an increase in the number of women who were identified with depressive symptoms in the UPHC after the screening started. Before the training, no women were identified with depressive symptoms as screening for mental health problems was not part of the routine procedures during antenatal visits in the UPHC. Following training, 25% of women were identified with depressive symptoms, which is consistent with the prevalence rate in LMIC (Maheshwari and Divakar, 2017; Roy and Chakma, 2017). The prevalence rate in our study is also similar to the pooled prevalence rate of perinatal depression (24.7%) reported in a meta-analytic study that analyzed data from 51 low- and middle-income countries (Roddy Mitchell et al., 2023). A meta-analytic study of antenatal depression in LMICs by Gelaye et al. (2016) also reports similar figures for one in four women. Early identification of mental health problems during pregnancy promotes the long-term well-being of the infant and the mother (Biaggi et al., 2016), and it is recommended that screening for mental health problems should be done routinely ("ACOG Committee Opinion No. 757", 2018). The majority of those identified with depressive symptoms had mild symptoms. The pregnant women identified with moderate depressive symptoms declined referral for further evaluation by a mental health professional citing fear of being perceived negatively for taking help for mental illness. This points to the need for education in the community addressing mental health disorders and stigma and providing intervention for women experiencing mild to moderate depression in the primary healthcare facility as well as integrating mental health services with routine antenatal care.

The study was conducted during the COVID-19 pandemic which likely limited the scope for training and for screening and may thus not be a true reflection of the scenario in a pandemic-free period without multiple restrictions. However, since the training and the screening could be conducted even during these challenging circumstances, it would likely be feasible to incorporate routine screening for mental health problems as part of antenatal care. In our training, we included healthcare workers from different professional and educational backgrounds, and this could have an effect on their knowledge, perceived skills and self-efficacy. However, this study also shows that the generic training we provided can be used by professionals from different backgrounds to make mental healthcare more accessible. Another limitation is that we could not ascertain whether no women with depressive symptoms had come for antenatal visits in the 3 months prior to training. However, since the prevalence rate of depressive symptoms is close to 25%, we assume that depressive symptoms were not checked by health workers in the UPHC. Continued training and supervision

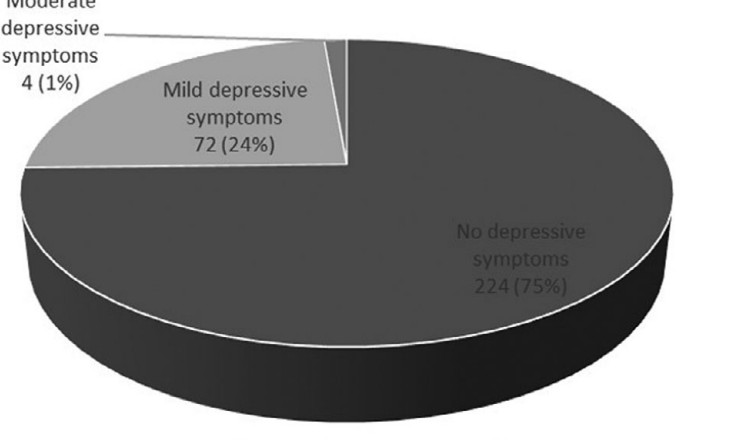

**Figure 2.** Proportion of subjects with depressive symptoms on screening after training as assessed by PHQ 9

are likely needed to sustain the effects of the intervention in the PHC.

In summary, our study showed that training can improve the knowledge, perceived skills and self-efficacy of nurses, nurse assistants and ASHAs. Incorporating screening for depressive symptoms into regular antenatal care as recommended by the ACOG ("ACOG Committee Opinion No. 757", 2018) is feasible in low-resource settings and increases the identification rate of depression. Policy decisions should be taken to incorporate mandatory screening for depression in antenatal care settings along with interventions based on illness severity (Prom et al., 2022) as it could lead to better care and reduction in the burden of mental health and improvement in health outcomes for both the mother and the child.

**Open peer review.** To view the open peer review materials for this article, please visit http://doi.org/10.1017/gmh.2023.48.

**Data availability statement.** Data will be available on request.

**Acknowledgements.** We thank Mr. Venkat Chekuri, Karuna Trust, for providing the necessary permissions and Dr. Archana Ashok for facilitating the study. We thank the medical officers and the Primary Health Center staff for their cooperation throughout the study period. We also thank Ms. Ashalata Ramthal and Ms. Jugnu for their assistance with data collection and entry.

**Author contribution.** S.T., M.E., T.T. and K.S. designed the study. S.T. and K.S. supervised data collection. T.T. and S.T. conducted the data analysis. The manuscript was prepared by S.T. All authors have read the manuscript and revised drafts and approved the final submission. All authors are accountable for all aspects of the work in ensuring that questions related to the accuracy or integrity of any part of the work are appropriately investigated and resolved.

**Financial support.** The study was supported by the Fogarty International Center of the National Institutes of Health under Award Number (D43TW009343) and the University of California Global Health Institute. Maria Ekstrand and Krishnamachari Srinivasan were supported by the US National Institute of Mental Health (R01MH100311).
The funding source had no role in the design of this study, its execution, analyses, interpretation of the data, or decision to submit results.

**Competing interest.** The authors declare that there is no conflict of interest.

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
