## [Reviewer Report]

I would like to submit the article titled ‘Feasibility of training primary health care workers to identify antenatal depression’ for consideration for publication in your esteemed journal. Our study examined the feasibility of providing training to primary health-care workers in an urban antenatal clinic in screening and identifying depressive symptoms in pregnant women. Our study provides insights into the feasibility of training health-workers in screening and identification of depression in antenatal care, especially in resource-poor countries where the burden of mental health care is high. We confirm that the manuscript has been submitted solely to this journal and is not published, in press, or submitted elsewhere.

---

## [Reviewer Report]

This is a good study that is potentially of benefit to practice in LMICs.

1. Please provide some more detail about Junior Health Assistants and Accredited Social Health Activists(ASHAs) since they formed the majority of the trainess and many potential readers of the article from outside the study country may not know who they are (their training/qualifications if any, what they do).

2. Related to the above, isnt it possible that one limitation of your findings which could impact on the generalizability especially in regard to your outcome findings of improvement in knowledge, perceived skills and behavioral efficacy could have been affected by the different levels of health training among the heterogenous cohort of the participants ?e.g it is plausible that nurses had more knowledge than Junior Health assistants and consequently skewed the findings. Please address this in your discussion and limitations

3.In your results and discussion you say “the results showed an increase in the number of women identified with depressive symptoms once a structured screening was initiated”. It is possible that there could have been no women with depressive symptoms in the three- month period prior to the training which formed your baseline prevalence. Secondly, did you only look at medical records for those clients seen by the study participants so that you could be able to reasonably conclude that they had not identified depresive symptoms prior to training?. For example it is possible that the pre-training records you looked at were of clients seen by different health care workers from the training participants. Please clarify this.

---

## [Reviewer Report]

This paper deals with task shifting in order to identify and provide treatment to pregnant women suffering from emotional difficulties. Given the high prevalence of these disorders in LMIC, task shifting is a must; but evidence needs to be gathered before the resources are allocated. So this research project addresses very valid questions. Unfortunately, it is difficult to read, particularly the Material and Methods and Discussion section because it is disorderly, with instruments described in the introduction and results reported in the discussion. I would respectfully ask the authors to rearrange these sections.

I have several specific comments as well.

1. Nurses and accredited social health activists are not comparable and they should not be described together, Nurses have had a specific training and so the contents and the outcome of knowledge based and skill training on them would be different from that given to ASHA’s. This is not to say ASHA’s cannot participate in screening. I strongly feel that the paper should report on Asha’s alone.

2. The impact statement reads like an alternative abstract. Please rephrase to emphasize the impact of this research .

Introduction

3. “The Patient Health Questionnaire (PHQ-9) is a 9-item, reliable and valid tool for identification of depression among pregnant women (Kroenke et al., 2001; Sidebottom et al., 2012)” page 2, para 2 doesn’t belong there but in Materials and methods

Material and Methods

4. Were nursing assistants trained also? No mention before.

Results and discussion

5. Page 9 last para: the prevalence of prenatal depression in LMIC is indeed around 25%. Your references report only on the prevalence in your region of the world. May I suggest you look at Gelaye 2016 and Mitchell 2023(JAMA Psychiatry. doi:10.1001/jamapsychiatry.2023.0069) for a more global review of prevalence?

6. “However, those with moderate symptoms declined referral for further evaluation by a mental health professional citing fear of being perceived negatively for taking help for mental illness.” (page 9, last para) I think this information belongs in Results, not in discussion. It is also a very controversial bit of information. If women who screen positive for depression refuse referral, what is the point of screening?

---

## [Reviewer Report]

To,

The Editor in Chief,

Cambridge Prisms: Global Mental Health

Dear Madam,

Thank you very much for reviewing the manuscript and for providing comments. I have responded to the comments and modified the manuscript. I am submitting two copies of the revised article titled ‘Feasibility of training primary health care workers to identify antenatal depression’ for consideration for publication in your esteemed journal. One copy has the modifications highlighted and the other is a clean copy. We confirm that the manuscript has been submitted solely to this journal and is not published, in press, or submitted elsewhere.

Thanking you, yours sincerely,

Dr. Susan Thomas

---

## [Reviewer Report]

Thank you for your responses and the subsequent edits made to the manuscript. I am satisfied that they have added more clarity and addresed my main concerns.